TOPICAL REVIEW

# Enhanced cycling of presynaptic vesicles during long-term potentiation in rat hippocampus

Kristen M. Harris 🆔

*Department of Neuroscience, Center for Learning and Memory, Institute for Neuroscience, University of Texas at Austin, Austin, TX, USA*

Handling Editors: Laura Bennet & Samuel Young

The peer review history is available in the Supporting Information section of this article (https://doi.org/10.1113/JP286983#support-information-section).

**Abstract figure legend** Illustration of the sequence of events leading to the sustained enhancement of presynaptic vesicle cycling during long-term potentiation (LTP). At time 0, all presynaptic axonal boutons contain a pool of non-docked vesicles and docked synaptic vesicles tethered to the presynaptic active zone. A particular presynaptic axonal bouton may or may not contain a mitochondrion or small dense core vesicle. Synapses comprise nascent zones with postsynaptic densities but no presynaptic vesicles, and active zones with both. The synaptic cleft spans both nascent and active zones. By 5 min after the induction of LTP, small dense core vesicles are recruited to the presynaptic membrane. Docked vesicles are reduced in number reflecting release. By 30 min, there are fewer vesicles overall, more coated pits, and small dense core vesicles are at their pre-LTP locations along the axons. In parallel, docked vesicles are recruited to regions of previous nascent zones converting them to active zones further enhancing the possibility of release. Two hours after the induction of LTP, the axonal bouton has enlarged and new nascent zones have appeared and are ready for new LTP. The docked vesicles are more tightly tethered and clustered at active zone release sites, suggesting a sustained elevation in presynaptic release during LTP.

The Journal of Physiology

**Abstract** Long-term potentiation (LTP) is a widely studied form of synaptic plasticity engaged during learning and memory. Here the ultrastructural evidence is reviewed that supports an elevated and sustained increase in the probability of vesicle release and recycling during LTP. In hippocampal area CA1, small dense-core vesicles and tethered synaptic vesicles are recruited to presynaptic boutons enlarging active zones. By 2 h during LTP, there is a sustained loss of vesicles, especially in presynaptic boutons containing mitochondria and clathrin-coated pits. This decrease in vesicles accompanies an enlargement of the presynaptic bouton, suggesting they supply membrane needed for the enlarged bouton surface area. The spatial relationship of vesicles to the active zone varies with functional status. Tightly docked vesicles contact the presynaptic membrane and are primed for release of neurotransmitter upon the next action potential. Loosely docked vesicles are located within 8 nm of the presynaptic membrane. Non-docked vesicles comprise recycling and reserve pools. Vesicles are tethered to the active zone via filaments composed of molecules engaged in docking and release processes. Electron tomography reveals clustering of docked vesicles at higher local densities in active zones after LTP. Furthermore, the tethering filaments on vesicles at the active zone are shorter, and their attachment sites are shifted closer to the active zone. These changes suggest more vesicles are docked, primed and ready for release. The findings provide strong ultrastructural evidence for a long-lasting increase in release probability following LTP.

(Received 18 November 2024; accepted after revision 14 February 2025; first published online 24 March 2025)

**Corresponding author** K. M. Harris: Department of Neuroscience, Centre for Learning and Memory, Institute for Neuroscience, University of Texas at Austin, Austin, TX 78712, USA. Email: kharris@utexas.edu

## Introduction

Long-term potentiation (LTP) is the persistent strengthening of synapses after a brief high-frequency stimulation and is widely accepted as a cellular correlate of learning and memory (Bliss & Gardner-Medwin, 1973; Nicoll, 2017). Within minutes after the induction of LTP, new receptors are inserted into the postsynaptic membrane. The resulting increase in the excitatory postsynaptic potential is immediate and can persist for hours *in vitro* or days to months *in vivo* (Bliss, 1993; Bliss & Gardner-Medwin, 1973; Bliss & Lomo, 1973; Capocchi et al., 1992; Larson & Lynch, 1986; Nguyen & Kandel, 1997; Staubli & Lynch, 1987). Quantal content is also increased soon after LTP induction and reflects an increase in the number of presynaptic vesicles that release neurotransmitter (Kullman & Nicoll, 1992; Liao et al., 1992; Malinow & Tsien, 1973; Stevens & Wang, 1994; Stricker et al., 1996). This increase in release probability is sustained several hours following LTP (Sokolov et al.,

2002), concurrent with postsynaptic growth and spine enlargement (Bourne & Harris, 2011). One might expect that the enhanced probability of release would involve increasing the number of vesicles docked and primed for neurotransmitter release. However, 2 h after induction of LTP, the total number of vesicles per presynaptic bouton is markedly decreased relative to boutons that received control stimulation only (Smith et al., 2016). These findings raise the question of whether an altered structure of docking and priming molecules leads to local clustering of vesicles that would elevate the probability of release following LTP. Here the ultrastructural evidence is reviewed that the presynaptic vesicle cycle is altered in a way consistent with elevated release and recycling that is sustained for at least 2 h after the induction of LTP.

## Homeostatic structural synaptic plasticity during LTP

Homeostatic structural synaptic plasticity has been demonstrated 2 h after the induction of LTP in the young

**Kristen M. Harris** earned her BS Summa Cum Laude in biology, chemistry and math at the Minnesota State University at Moorhead, her MS in Neurobiology at the University of Illinois, and her PhD in Neurobiology at Northeastern Ohio Universities College of Medice and Kent State University. She completed her postdoctoral work in the department of neurology at the Massachusetts General Hospital and was recruited to and rose through the ranks to Associate Professor in the departments of neuropathology and neurology at Harvard Medical School. She was then recruited as tenured Professor to Boston University, then as a Georgia Research Alliance Eminent Scholar and professor at the Medical College of Georgia, and since 2006 as Professor in the Department of Neuroscience and Center for Learning and Memory at the University of Texas at Austin. She also holds a position as adjunct professor at the Salk Institute for Biology and is a member of the National Academy of Science.

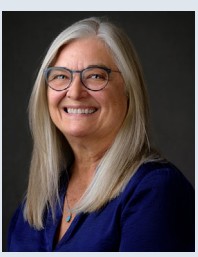

adult rat hippocampus (Bourne & Harris, 2011; Bell et al., 2014). In this work, LTP was induced in hippocampal slices by a saturating theta-burst stimulation delivered to one electrode while only test pulses were delivered to a separate control electrode in the same slice (Fig. 1*A* and *B*). Slices were then fixed in mixed aldehydes using our rapid microwave-enhanced protocol to produce high quality preservation of synapses (Fig. 1*C*). The synaptic surface area was estimated by measuring the region of the axon–spine interface occupied by the postsynaptic density (PSD). The size of the PSD areas remained constant across the first 30 min after induction of LTP but was significantly enlarged by 2 h (Fig. 1*D*). In contrast, LTP stalls the small spine outgrowth resulting from control stimulation (Fig. 1*E*). The density of large spines was not altered by control or LTP conditions, and thus synapse enlargement is likely restricted to existing large spines (Chirillo et al., 2019). Consequently, the total PSD area supported per unit length of dendrite remained constant after LTP (Fig. 1*F*). Together these findings illustrate a homeostatic balance in total synaptic weight and control activation stimulates spine outgrowth while LTP enlarges synapses at the expense of new spine formation in adult animals (Fig. 1*G*).

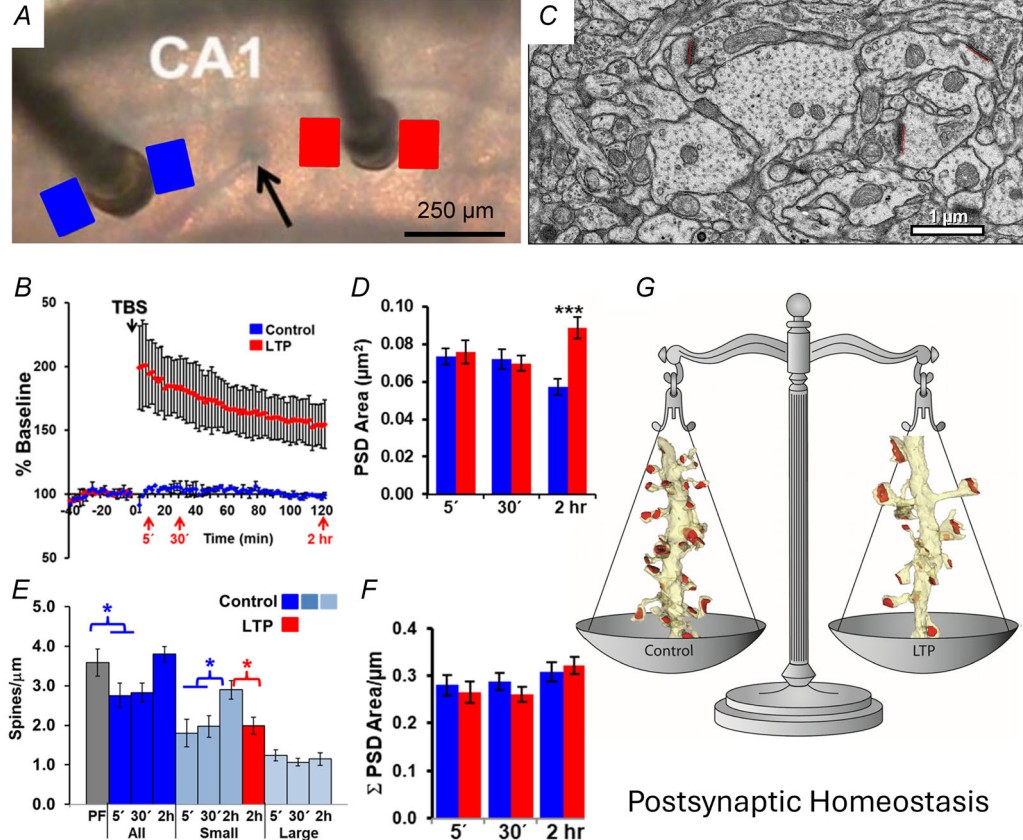

**Figure 1. Homeostatic synaptic plasticity during LTP in the young adult rat hippocampus**
*A*, positioning of stimulating electrodes and sample locations where serial sections were obtained for electron microscopy (EM) from the control (blue) and LTP (red) sites. The stimulating electrode that induced LTP was alternated between the CA3 (left) and the subicular (right) side of the recording electrode (black arrow). *B*, LTP was induced by theta-burst stimulation (TBS; 8 trains at 30 s intervals of 10 bursts per train at 5 Hz with 4 pulses in each burst at 100 Hz), and the control side received test pulses at the same rate as LTP test pulses of 1 per 2 min over the course of the experiment. *C*, illustration of the excellent ultrastructure from an adult hippocampal slice using our rapid microwave-enhanced fixation protocol (Jensen & Harris, 1989). This example is from a P61 rat hippocampus in the 30 min control condition. Red lines indicate example postsynaptic densities (PSD). *D*, the average size of PSD areas of synapses enlarged during LTP; (*E*) while the increase in small-spine density seen under control conditions was absent during LTP. *F*, consequently, the total PSD area after LTP induction equalled that in control conditions. *G*, together these findings resulted in a homeostatic balance in total synaptic weight per unit length of dendrite. *A–D*, *F* and *G*, are adapted from Bourne & Harris (2011); *E* is adapted from Bell et al. (2014). *$P < 0.05$; ***$P < 0.001$.

## Changes in presynaptic boutons during LTP

Synapses on dendritic spines in s. radiatum of hippocampal area CA1 comprise Schaffer collateral axons from the ipsilateral area CA3, commissural axons from the contralateral CA3 axons, and longitudinal projections from neighbouring components of the ipsilateral hippocampus. Three-dimensional reconstructions illustrate the non-parallel trajectories of these axons (Fig. 2*A* and *B*) that give rise to stimulation of independent

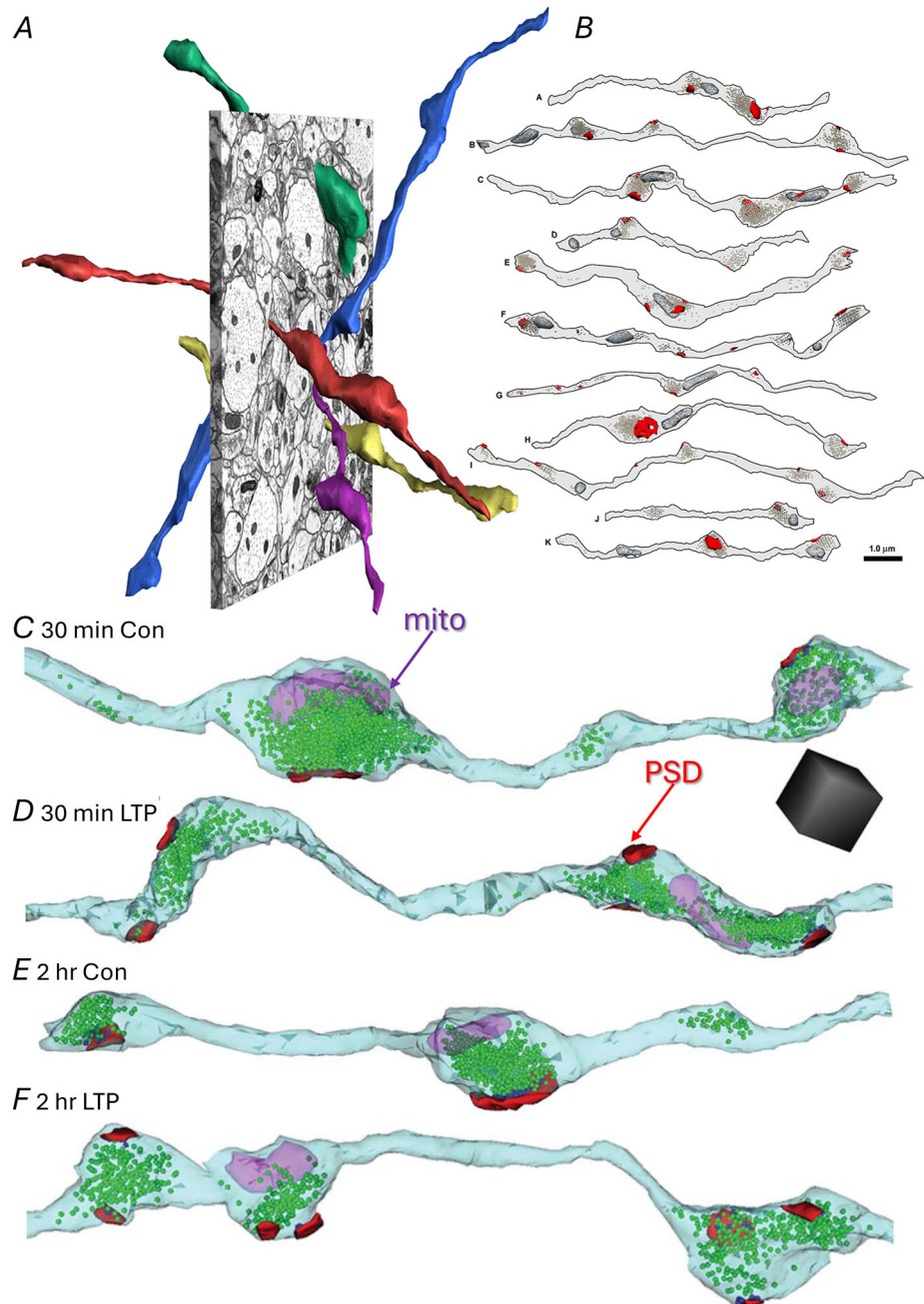

**Figure 2. CA3 axons that synapse in s. radiatum of hippocampal area CA1**
*A*, central EM and 3D reconstructions of these axons illustrating their non-parallel trajectories that give rise to stimulation of independent synapses at a separation between the electrodes greater than 200 μm using the protocol shown in Fig. 1. *B*, illustration showing that only 50% of the presynaptic boutons along these axons contain mitochondria (dark grey structures) in the perfusion-fixed hippocampus, *in vivo*. (*A* and *B* are adapted from Shepherd & Harris, 1998). *C–F*, reconstructed axons from the control (Con) conditions (*C*, *E*) and 30 min (*D*) or 2 h (*F*) after induction of LTP in the same slice. Axon, pale blue; vesicles, green spheres; mitochondria (mito), fuchsia; PSDs, red. Adapted from Bourne et al. (2013).

synapses when the two stimulating electrodes are separated by more than 200 µm (Bourne et al., 2013; Shepherd & Harris, 1998; Smith et al., 2016). Only about 50% of the presynaptic boutons along these axons contain mitochondria in control or LTP conditions (Fig. 2*C–F*) (Bourne et al., 2013; Smith et al., 2016). Related to the stalled spine outgrowth, a specific population of single synaptic boutons is lower after LTP (Fig. 3*A*), whereas the frequency of multisynaptic (Fig. 3*B*) and non-synaptic (Fig. 3*C*) boutons is not altered after LTP relative to control stimulation (Fig. 3*D–F*). These outcomes suggest that multisynaptic boutons are more stable, and that the spines are not lost from pre-existing single synaptic boutons. Instead, the stalled spine outgrowth results in the formation of fewer presynaptic boutons during LTP in adult hippocampal slices. The outcomes during development are different in that LTP promotes new spine outgrowth and related presynaptic plasticity (Ostroff et al., 2018; Smith et al., 2016; Watson et al., 2016). That is a story for other reviews (Harris, 2020a, b). Here we will focus on the outcomes in young adult hippocampus.

## Sustained decrease in the number of presynaptic vesicles during LTP

In addition to the loss of presynaptic boutons, there is a substantial drop in the number of vesicles found in the remaining boutons (Bourne et al., 2013). This over-all reduction affects both the docked and non-docked vesicle pools at 30 min and is sustained in the non-docked vesicle pool for at least 2 h (Fig. 4). The drop in docked and non-docked vesicle number is greatest in boutons that contain clathrin-coated pits, suggesting there is an elevated release and recycling of vesicles during LTP (Fig. 5). Furthermore, the drop in presynaptic vesicles is greatest in presynaptic boutons that contain a mitochondrion, suggesting an elevated requirement for ATP and local regulation of calcium in the boutons with

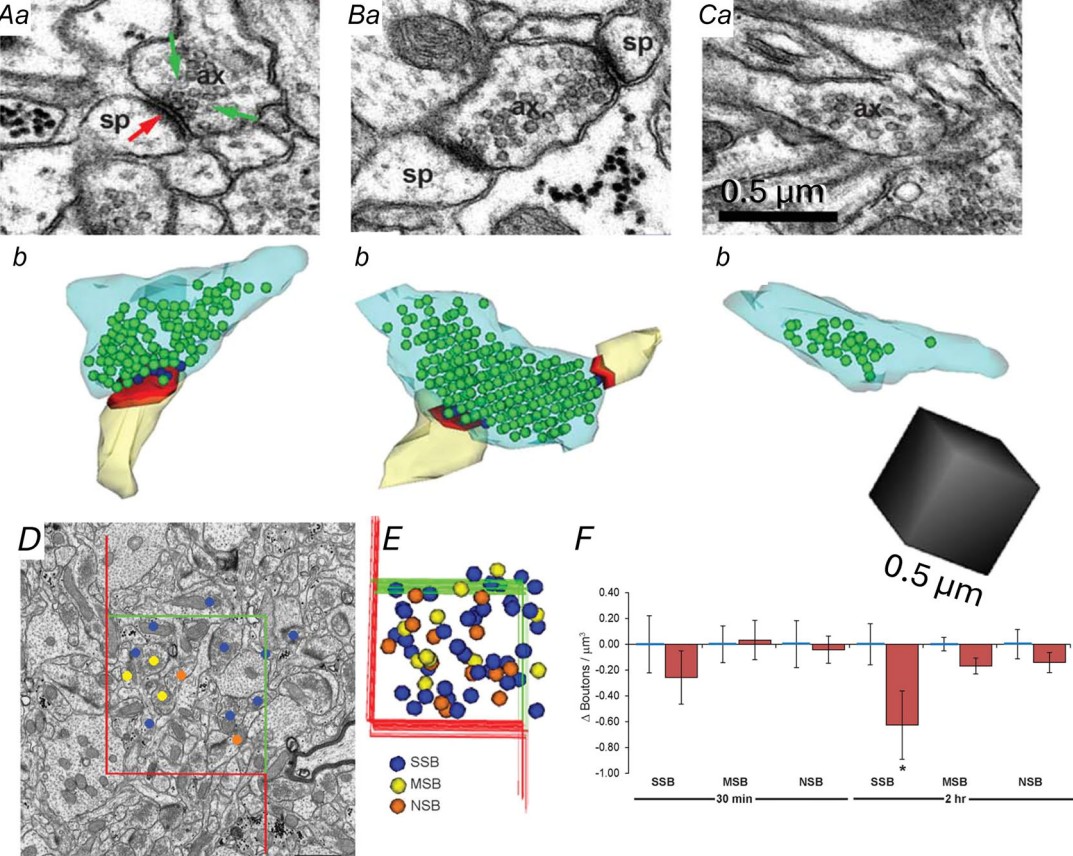

**Figure 3. The stalled spine outgrowth results in fewer single synaptic boutons (SSBs)**
*Aa–Ca*, EM and *b*, 3D reconstructions of SSB *A*, multisynaptic bouton (MSB) *B*, and non-synaptic bouton (NSB) (*C*). ax, axon; sp, spine. *D*, central EM and *E*, sampling brick across serial EM volume to identify and include boutons contained within or touching the green surfaces while excluding those touching the red surface. *F*, brick analyses revealed no differences at 30 min, but fewer SSBs at 2 h after induction of LTP ($P < 0.05$) with no significant changes in MSBs or NSBs. Scale bar in *Ca* is for *Aa–Ca* and cube in *Cb* is for *Ab–Cb* at 0.5 µm per side, 0.125 µm³.

enhanced vesicular release after LTP (Fig. 6; Smith et al., 2016).

## Vesicle filling of presynaptic nascent zones enlarges the active zone during LTP

The ultrastructure of the presynaptic active zone is defined by the location where vesicles dock and release neurotransmitter (Neher & Brose, 2018). For excitatory synapses on dendritic spines in the mammalian brain, this region is associated with the PSD (Fig. 7*A*). There is another region defined by the presence of the PSD but absence of presynaptic vesicles that we term the nascent zone (Fig. 7*B–E*). Presynaptic vesicles are tethered to small dense core vesicles in the CA1 excitatory axons (Fig. 7*F* and *G*) (Sorra et al., 2006). Within 5 min after the induction of LTP in hippocampal area CA1, small dense core vesicles are recruited to the presynaptic bouton (Fig. 7*F* and *G*). Occasionally, the small dense core vesicles can also be found docked to the edges of the active zone area, where they could guide the tethered vesicles to the correct location to enlarge the active zone (Fig. 7*H*). The dark core of the small dense

core vesicles contains cell adhesion molecules, including neurexin, which upon release would anchor the post-synaptic glutamatergic receptors (AMPA) to the previous nascent zone area, thereby creating nanocolumns in the right position to enlarge the active zone (Haas et al., 2018; Harris et al., 2024).

Importantly, the surface of the dense core vesicle membrane contains the scaffolding proteins (piccolo and bassoon) needed to tether presynaptic vesicles to docking sites (Dresbach et al., 2006; Zhai et al., 2001; Ziv & Garner, 2004). Interestingly, the dimensions of the membrane surface area of the small dense core vesicles would be sufficient to convert an entire nascent zone to an active zone (Fig. 7*I*). By 30 min after the induction of LTP, the amount of the postsynaptic density attributed to nascent zones decreases (Fig. 7*J*), suggesting that the vesicles recruited by the small dense core vesicles could have served immediately to enlarge the presynaptic active zone, and enhance the probability of release. Thus, the subsequent enlargement of the postsynaptic density described in Fig. 1*D* above is silent because that growth is primarily attributable to the addition of nascent zones, having no presynaptic vesicles (Fig. 7*K* and *L*).

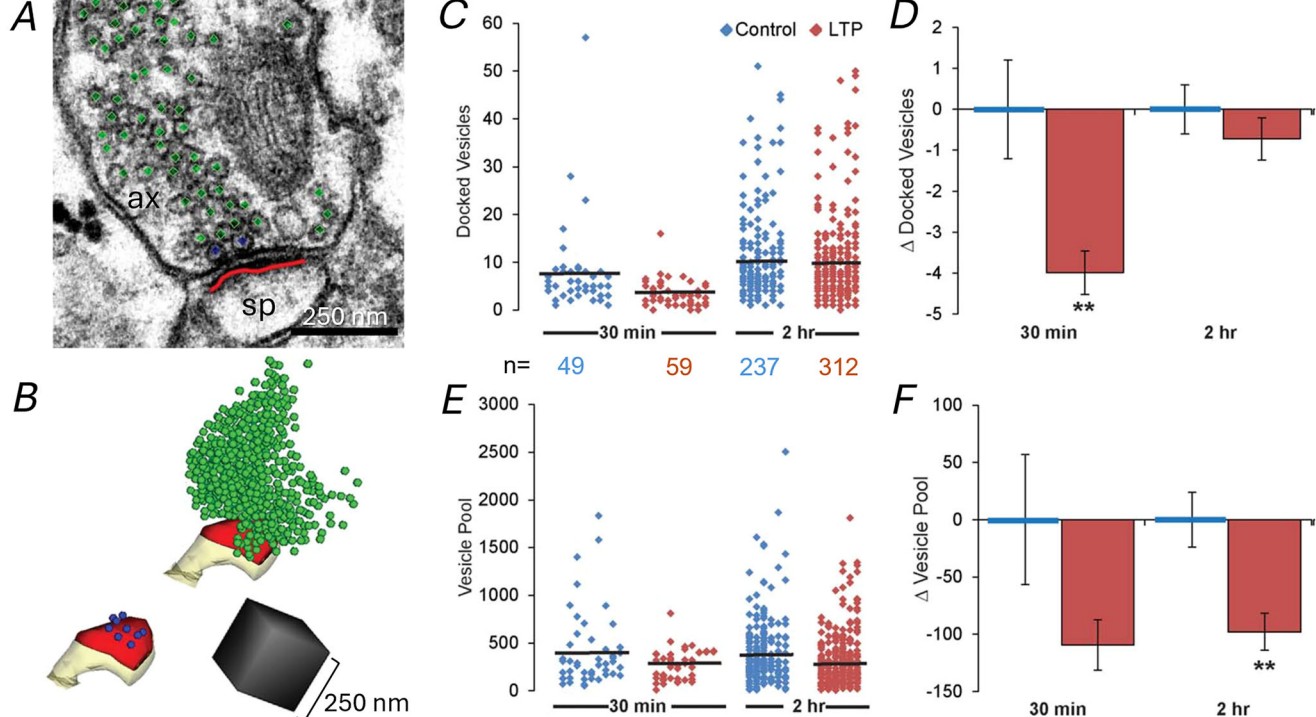

**Figure 4. Sustained decrease in number of presynaptic vesicles during LTP**
*A*, EM and *B*, 3D reconstruction of a dendritic spine (sp, yellow) and presynaptic bouton (ax) with the docked vesicles (blue), vesicles in the pool (green), and PSD (red). *C*, average number of docked vesicles for each condition (black lines) with distribution from individual boutons (*n* = number of boutons in each condition). *D*, decrease in the number of docked vesicles per presynaptic bouton at 30 min after induction of LTP relative to control values (*P* < 0.01). *E*, average number of non-docked vesicles (black lines) for each condition with distribution from individual boutons. *F*, by 2 h after LTP induction, the vesicle pool was significantly smaller relative to control (*P* < 0.01).

## Increased local density of tightly docked vesicles in active zones during LTP

We wondered how to reconcile the overall decrease in presynaptic vesicles with the enhanced probability of release associated with LTP. To answer this question, regions of presynaptic active zones that had at least one docked vesicle were targeted for visualization with electron tomography (Jung et al., 2021). The exact position of vesicles in the active zone varies with functional status (Neher & Brose, 2018). Tightly docked vesicles touch the presynaptic membrane and are primed for the release of neurotransmitter. Loosely docked vesicles (less than 8 nm) and non-docked vesicles (greater than 8 nm) comprise recycling and reserve pools. Electron tomography reveals precise positions of the docked presynaptic vesicles and their filamentous tethers composed of molecules involved in docking, priming and release (Neher & Brose, 2018). Tightly docked vesicles are clustered producing local

increases in their density during LTP (Fig. 8*A*–*F*), whereas the loose or non-docked vesicle densities (Fig. 8*G*–*J*) are unchanged relative to control stimulation (Fig. 8*K*). All vesicles congregating within 45 nm above the active zone have tethering filaments attached to the presynaptic membrane (Fig. 8*C*, *D* and *L*–*O*). The tethering filaments are shorter for both tight and loosely docked vesicles after LTP (Fig. 8*P*). In addition, the vesicle attachment sites of the tethering filaments shift downward towards the presynaptic membrane and horizontally towards the docking site resulting in shorter filaments. These alterations would stabilize docked vesicles at the active zone and facilitate formation of SNARE complexes. Shortened tethering filaments on the loosely or non-docked vesicles would enhance recruitment of vesicles to the docking sites at the active zone and increase their readiness for release (Cole & Reese, 2023; Cole et al., 2016b). In summary, the shortening of tethering filaments is likely stabilizing more vesicles in the primed state long after the induction

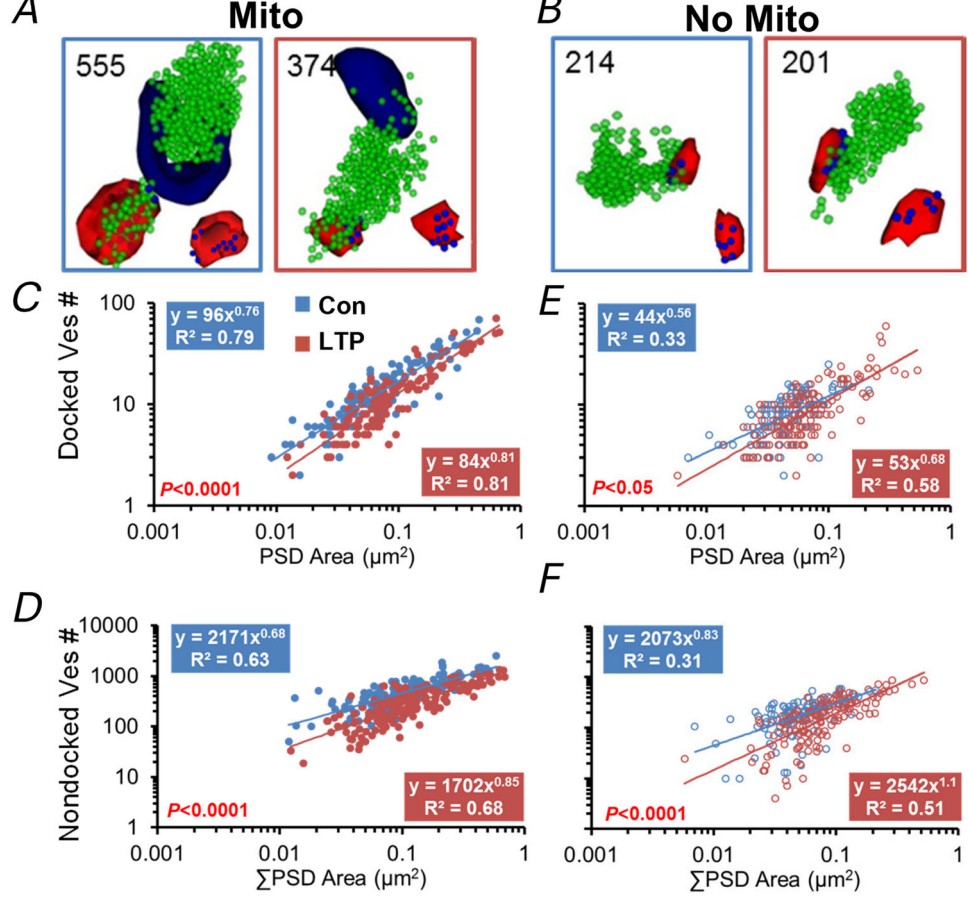

**Figure 5. Vesicle drop during LTP is greatest in boutons containing a mitochondrion**
*A* and *B*, representative vesicle composition in presynaptic boutons (*A*) containing mitochondria (dark blue) or (*B*) without mitochondria from the control and LTP conditions. *C*–*F*, both docked and non-docked vesicles had greater drops across synapses of all PSD areas for boutons with mitochondria (*C*, *D*) versus no mitochondrion (*E*, *F*). Adapted from Smith et al. (2016).

of LTP. Primed vesicles comprise the readily releasable pool, releasing neurotransmitter upon the next action potential. Hence, stabilizing vesicles in the primed state could contribute to the enhanced probability of release occurring several hours after LTP induction.

## Involvement of presynaptic vesicles in axon building

When Heuser and Reese first discovered that presynaptic vesicles are recycled locally at the neuromuscular junction, they opened the field to learn how axons throughout the nervous system regulate synaptic transmission. In

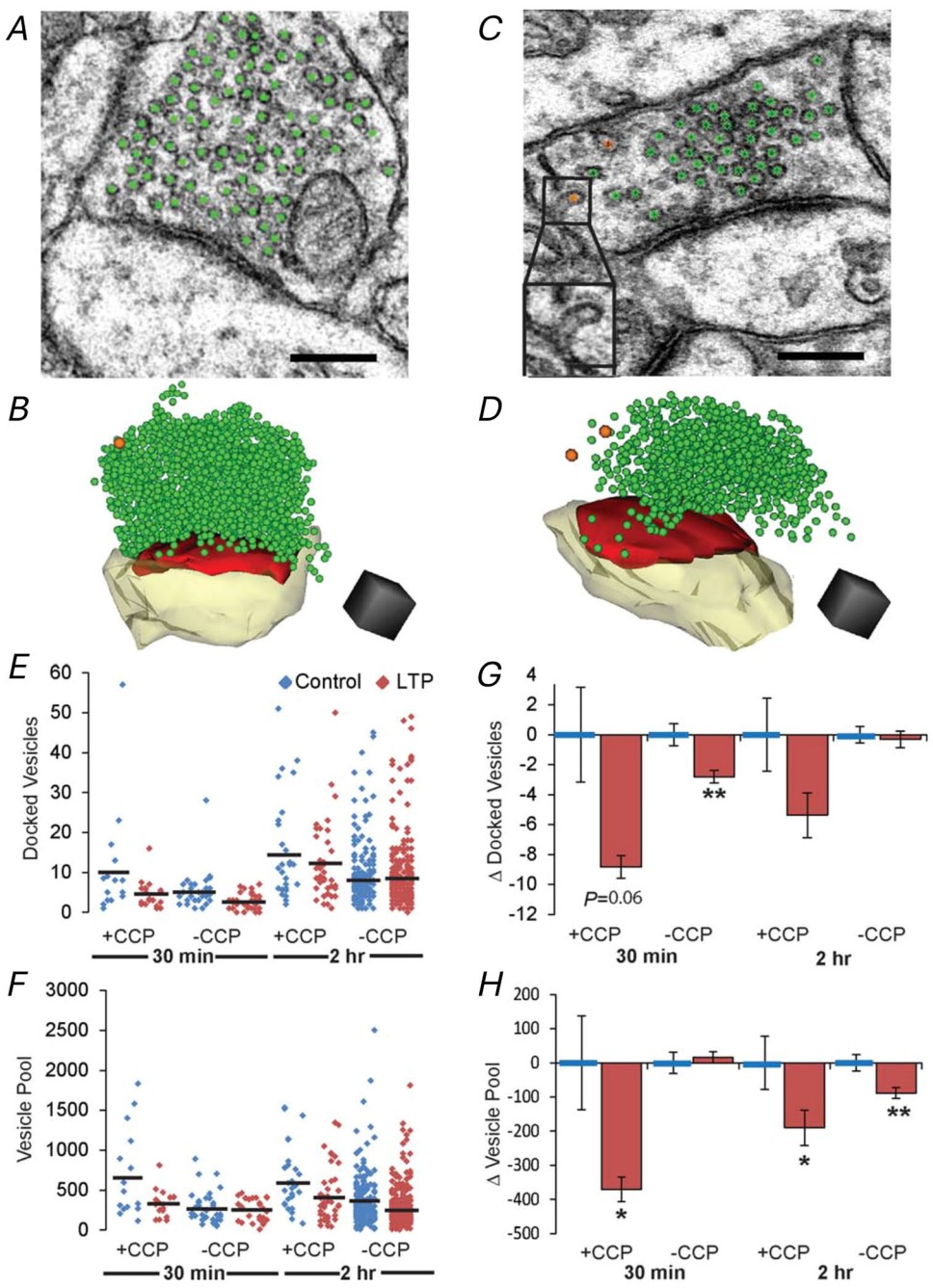

**Figure 6. The number of vesicles was substantially reduced in boutons containing clathrin coated pits (CCP)**
*A–D*, EM and 3D reconstructions of presynaptic vesicles (green), CCPs (orange), PSDs (red) and spines (yellow) from control (*A*, *B*) and 2 h LTP (*C*, *D*) conditions. *E* and *F*, graphs of vesicle counts in presynaptic boutons with and without one or more CCPs. *G* and *H*, statistically significant reductions (*$P < 0.05$, **$P < 0.01$) in docked vesicles and vesicle pool numbers in boutons with (+CPP) or without (–CCP) one or more CCPs.

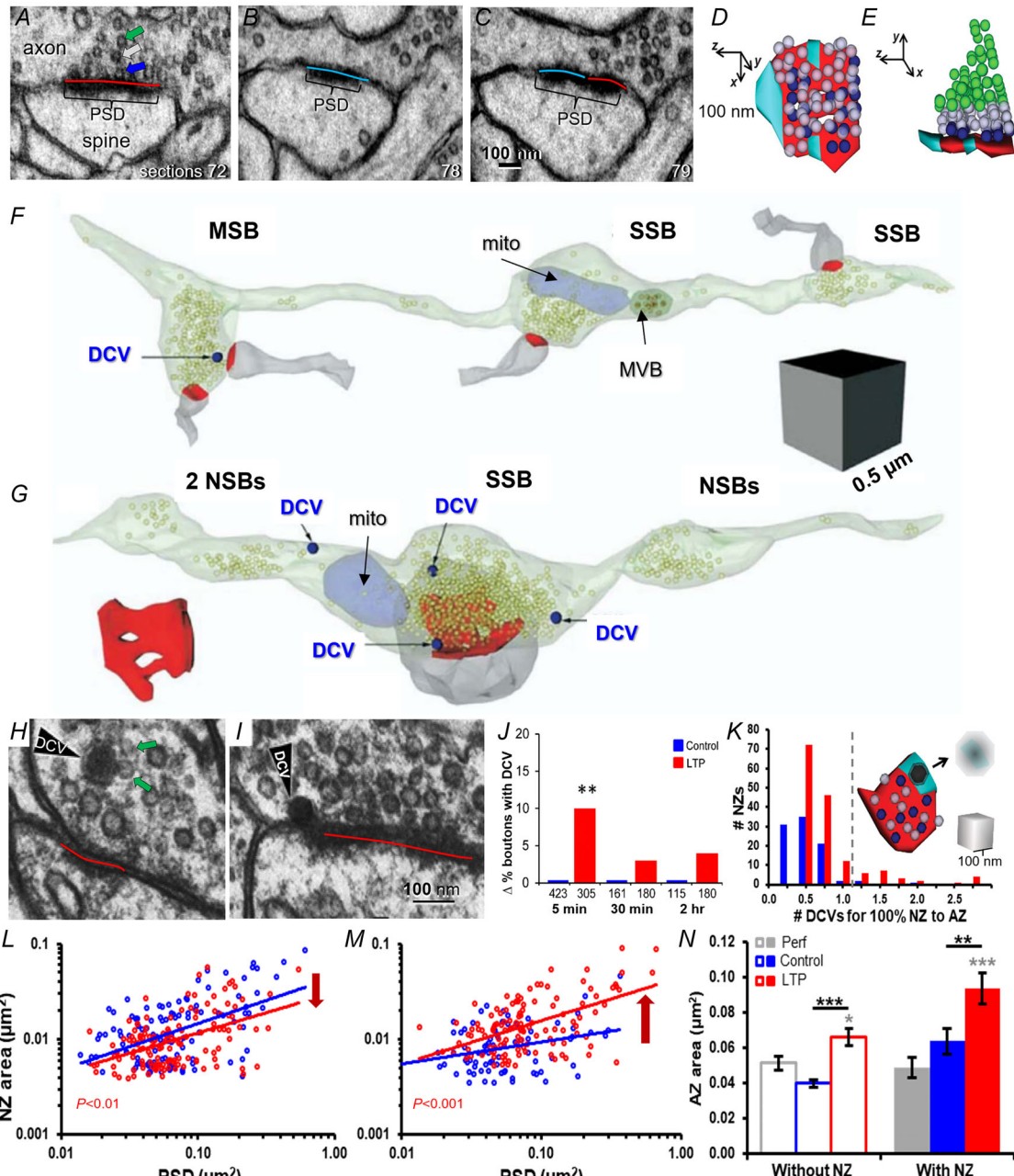

**Figure 7. Small dense core vesicles and the recruitment of presynaptic vesicles to convert nascent zones to active zones during LTP**

*A*, active zone (AZ, red) with docked (royal blue arrow), non-docked (white arrow), and reserve pool (green arrow) vesicles, at a PSD (red). *B* and *C*, nascent zone (NZ, aqua) has a PSD but no presynaptic vesicles. *D* and *E*, 3D reconstructions of the synapse illustrated in *A*–*C*. *F* and *G*, 3D reconstructions of axons in s. radiatum of hippocampal area CA1 illustrating their composition of synaptic vesicles (light green spheres), dense core vesicles (DCV), mitochondria (mito), and multivesicular body (MVB). These reconstructions are from perfusion fixed brain and illustrate neighbouring single-synaptic (SSB), multisynaptic (MSB), and non-synaptic (NSB) boutons along single axons. *H*, EM illustrating synaptic vesicles that are tethered (green arrows) to (*I*) a dense core vesicle (DCV). *I*, DCV at the edge of an AZ. *J* DCVs are recruited from inter-bouton regions to synaptic boutons at 5 min after TBS (*n* = number of synapses). *K*, plot of the number of DCVs that would be needed to convert a NZ to AZ by filling it with the tethered docked vesicles *versus* the number of NZs in the control or LTP conditions that would be fully converted to AZ if a DCV were to be recruited with tethered vesicles. *L*, NZ area decreases by 30 min after LTP induction. *M*, NZ size is re-elevated by 2 h after LTP induction. *N*, NZ recovery at 2 h during LTP is greatest on spines with the largest AZ areas (Perf: *in vivo* controls). (Adapted from Bell et al., 2014; Harris et al., 2024; Sorra et al., 2006.)

hippocampal area CA1 we have learned that the presynaptic axonal boutons are highly diverse, ranging in total vesicle number from a few vesicles to more than 2000 per bouton. Furthermore, the number of docked vesicles per bouton is also highly variable ranging from a few to more than 50 per active zone. Unlike the neuromuscular junction, where virtually all presynaptic boutons contain local mitochondria, only about 50% of presynaptic axonal boutons contain mitochondria in area CA1 s. radiatum (Shepherd & Harris, 1998; Smith et al., 2016). Consistent with the vesicle hypothesis, there is an elevation in clathrin-coated pits and vesicles in boutons with the greatest drop in total vesicle number after LTP, an effect that subsides with time. However, the vesicle cycle takes about 1 min from release to recycling (Sudhof, 1995, 2004). Hence, it appears that there are hundreds to thousands of extra vesicles in many of the presynaptic boutons that are not necessarily part of the release or recycling pool. Evidence is accumulating that the presynaptic boutons are enlarged after LTP (Chereau et al., 2017), in proportion to the amount of membrane that would be contained in the lost pool of presynaptic vesicles during LTP (pre-published observations, Garcia et al., 2024). Perhaps not all vesicles are engaged in release and recycling, but some instead provide the membrane needed for presynaptic bouton enlargement during LTP.

## Relevance of aldehyde fixation

In the past, rapid freezing and freeze-substitution have been used to capture vesicle dynamics during synaptic transmission (Cole et al., 2016a; Heuser et al., 1979; Imig et al., 2014; Watanabe et al., 2013, 2014; Zampighi et al., 2008, 2014). The ultrastructural synaptic plasticity reviewed here was obtained from microwave-enhanced aldehyde fixation of brain slices that likely occurs over seconds rather than the milliseconds achieved with freezing (Jensen & Harris, 1989). In cultured hippocampal neurons, aldehyde fixation does not alter the probability of vesicular release (Rosenmund & Stevens, 1997). Tethering filaments are preserved in both aldehyde-fixed and rapid frozen synapses (Cole, & Reese, 2023; Cole et al., 2016b; Harlow et al., 2001; Jung et al., 2016). Relative to perfusion fixation *in vivo*, the effects of slicing on presynaptic release are resolved within an hour of incubation *in vitro* (Fiala et al., 2003), while the LTP effects were measured after approximately 6 h *in vitro*. Any other effects related to the time *in vitro* were controlled for by test pulse stimulation in the same slice. Hence, we conclude that these observations faithfully reflect structural synaptic plasticity that underlies a sustained elevation in the probability of release at 2 h after the induction of LTP.

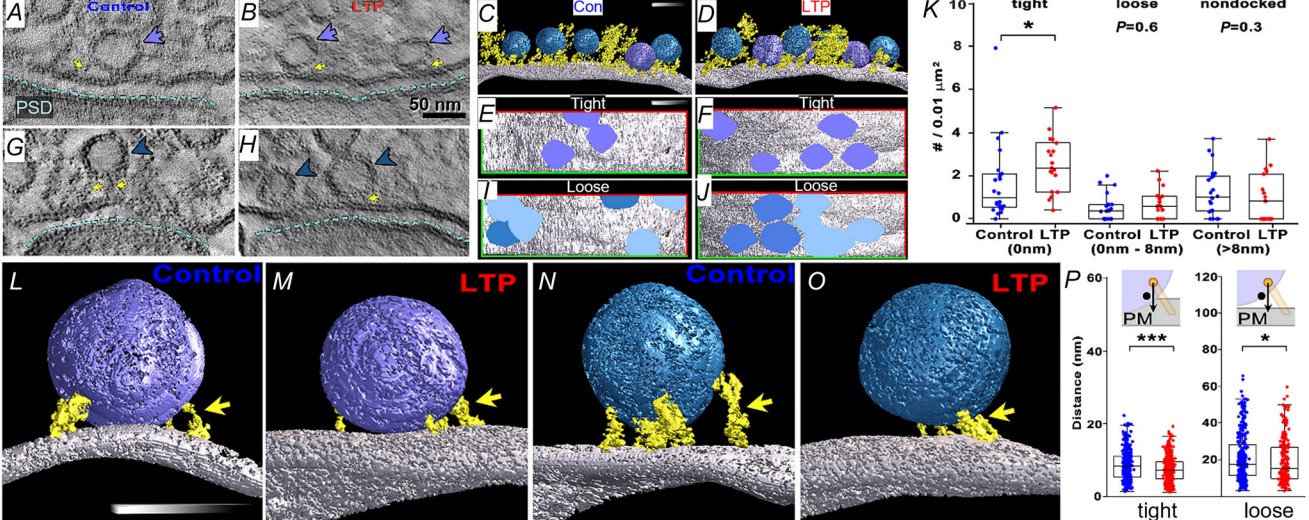

**Figure 8. Effects of LTP on the local density of tethered and docked presynaptic vesicles**
*A–J*, tightly docked (purple), loosely docked (turquoise) or non-docked (light blue) vesicles with molecular tethering filaments (yellow). *C* and *D*, side view of three-dimensional tomogram illustrates presynaptic membrane surface (silver) with tight and loose docked vesicles and tethering filaments. *E–J*, unbiased sampling frames to compute local cluster densities for tight (*E*, *F*) and loose or non-docked (*I*, *J*) vesicles. *K*, the density of tightly docked vesicles, but not loosely or non-docked vesicles, increases in local clusters after LTP. *L–P*, the length of the tethering filaments (yellow) for both tight (*L*, *M*) and loosely docked (*N*, *P*) vesicles was shortened at 2 h after LTP induction. (Adapted from Harris et al., 2024; Jung et al., 2021.)

## Future tomography in large image volumes

Current EM tomographic field size limits investigation to small parts of the synapse. Presynaptic mitochondria support greater vesicular release at tonic versus phasic synapses (Brodin et al., 1999). Mitochondria-containing boutons sustain a greater loss of synaptic vesicles during LTP than boutons lacking mitochondria (Smith et al., 2016). None of our EM tomographic volumes are currently large enough to capture presynaptic mitochondria, hence the outcomes reviewed here likely include boutons with and without mitochondria diluting mitochondria-specific effects. Combining EM tomography with wide field scanning electron microscopy is needed to elucidate nanoscale effects of synaptic plasticity across whole synapses and networks (Kuwajima et al., 2013). Similarly, extending this work to synapses of all types in local and broad networks will require tomography at a connectomics scale to understand the time course and network effects of altered presynaptic vesicular release and recycling during learning, memory and disease states that disrupt normal function.

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

## Additional information

### Competing interests

The authors declare they have no competing interests.

### Author contributions

Sole author.

### Funding

The author is funded by the National Institute of Mental Health (NIMH), R01MH095980, and the NSF Directorate for Biological Sciences (BIO), NeuroNex2014862.

## Keywords

axon, dendrite, nanoscale, spines synapse, synaptic plasticity, ultrastructure

## Supporting information

Additional supporting information can be found online in the Supporting Information section at the end of the HTML view of the article. Supporting information files available:

**Peer Review History**

