## [Peer Review History · The Journal of Physiology]

ENHANCED CYCLING OF PRESYNAPTIC VESICLES DURING LONG-TERM POTENTIATION (LTP) IN RAT HIPPOCAMPUS

Kristen M. Harris

DOI: 10.1113/JP286983

Corresponding author(s): Kristen Harris (kharris@utexas.edu)

The following individual(s) involved in review of this submission have agreed to reveal their identity: Noa Lipstein (Referee #2)

Review Timeline:

Submission Date:	18-Nov-2024
Editorial Decision:	11-Dec-2024
Revision Received:	20-Jan-2025
Editorial Decision:	28-Jan-2025
Revision Received:	13-Feb-2025
Accepted:	14-Feb-2025

Senior Editor: Laura Bennet

Reviewing Editor: Samuel Young

Transaction Report:

Dear Dr Harris,

Re: JP-TR-2024-286983 "ENHANCED CYCLING OF PRESYNAPTIC VESICLES DURING LTP IN RAT HIPPOCAMPUS"
by Kristen M. Harris

Thank you for submitting your manuscript to The Journal of Physiology. It has been assessed by a Reviewing Editor and by 2 expert referees and we are pleased to tell you that it is potentially acceptable for publication following satisfactory major revision.

Please address all the points raised and incorporate all requested revisions or explain in your Response to Referees why a change has not been made. We hope you will find the comments helpful and that you will be able to return your revised manuscript within 2 months. If you require longer than this, please contact journal staff: jp@physoc.org. Please note that this letter does not constitute a guarantee for acceptance of your revised manuscript.

ABSTRACT FIGURES: Authors are expected to use The Journal's premium BioRender account to create/redraw their Abstract Figures. Information on how to access this account is here:

<https://physoc.onlinelibrary.wiley.com/journal/14697793/biorender-access>.

REVISION CHECKLIST:

IMPORTANT POINTS TO NOTE WHEN REVISING YOUR MANUSCRIPT:

We look forward to receiving your revised submission.

Yours sincerely,

Laura Bennet
Senior Editor
The Journal of Physiology

REQUIRED ITEMS

- Please include an Abstract Figure file, as well as the Figure Legend text within the main article file. The Abstract Figure is a piece of artwork designed to give readers an immediate understanding of the Review Article and should summarise the main conclusions. If possible, the image should be easily 'readable' from left to right or top to bottom. It should show the physiological relevance of the Review so readers can assess the importance and content of the article. Abstract Figures should not merely recapitulate other figures in the Review. Please try to keep the diagram as simple as possible and without superfluous information that may distract from the main conclusion of the Review. Abstract Figures must be provided by authors no later than the revised manuscript stage and should be uploaded as a separate file during online submission labelled as File Type 'Abstract Figure'. Please ensure that you include the figure legend in the main article file. All Abstract Figures will be sent to a professional illustrator for redrawing and you may be asked to approve the redrawn figure before your paper is accepted.

- Your MS must include a complete "Additional information section" with the following 4 headings and content:

Competing Interests: A statement regarding competing interests. If there are no competing interests, a statement to this effect must be included. All authors should disclose any conflict of interest in accordance with journal policy.

Author contributions: Each author should take responsibility for a particular section of the study and have contributed to writing the paper. Acquisition of funding, administrative support or the collection of data alone does not justify authorship; these contributions to the study should be listed in the Acknowledgements. Additional information such as 'X and Y have contributed equally to this work' may be added as a footnote on the title page.

It must be stated that all authors approved the final version of the manuscript and that all persons designated as authors qualify for authorship, and all those who qualify for authorship are listed.

Funding: Authors must indicate all sources of funding, including grant numbers. If authors have not received funding, this must be stated.

It is the responsibility of authors funded by RCUK to adhere to their policy regarding funding sources and underlying research material. The policy requires funding information to be included within the acknowledgement section of a paper. Guidance on how to acknowledge funding information is provided by the Research Information Network. The policy also requires all research papers, if applicable, to include a statement on how any underlying research materials, such as data, samples or models, can be accessed. However, the policy does not require that the data must be made open. If there are considered to be good or compelling reasons to protect access to the data, for example commercial confidentiality or legitimate sensitivities around data derived from potentially identifiable human participants, these should be included in the statement.

Acknowledgements: Acknowledgements should be the minimum consistent with courtesy. The wording of acknowledgements of scientific assistance or advice must have been seen and approved by the persons concerned. This section should not include details of funding.

- The reference list must be in alphabetical order, rather than numbered, to comply with our Journal format.

- Please upload separate high quality figure files via the submission form.

- Author profile(s) must be uploaded via the submission form. Authors should submit a short biography (no more than 100

words for one author or 150 words in total for two authors) and a portrait photograph of the two leading authors on the paper. These should be uploaded and clearly labelled together in a Word document with the revised version of the manuscript. Any standard image format for the photograph is acceptable, but the resolution should be at least 300 DPI and preferably more. A group photograph of all authors is also acceptable, providing the biography for the whole group does not exceed 150 words.

- It is the authors' responsibility to obtain any necessary permissions to reproduce previously published material and to list these within the main article file. For information, please see: https://jp.msubmit.net/cgi-bin/main.plex?form_type=display_requirements#permissions.

- Please ensure that the Article File you upload is a Word file.

EDITOR COMMENTS

Reviewing Editor:

This manuscript focuses on the synaptic ultrastructural changes associated with LTP in the hippocampus. Both reviewers agreed that the topic is timely and addressed an important topic in the field. However, both reviewers had issues with the review. There was significant concern that this was not really a review at all, but more of a perspective of the author's own work. Therefore, to give a more representative review of this important topic, the author should provide more of a broader review of the field that takes into account of other contributors to the field. This can be addressed with rewriting the text based on the positive comments and critiques by both reviewers.

Please also see 'Required Items' above.

REFEREE COMMENTS

Referee #1:

This is a perspective article from a renowned expert on synapse structure and function, focusing on the presynaptic ultrastructural changes associated with long-term potentiation (LTP) in the hippocampus. The article takes the reader through series of experiments drawn from her own work in support for enhanced presynaptic vesicle recycling during LTP. A sweeping summary of key observations is informative and of general interest. The article as it stands is somewhat convoluted and following are some suggestions to enhance accessibility.

- The author should clarify what the article is about at the outset. While it is mentioned as a "review", it is not a typical review of a topic but rather a review of one's own work. This comes as a surprise while reading the article.

- Relating to the point above, perhaps a broader introduction including some previous work from others that have implicated presynaptic ultrastructural changes in LTP should be included, for example, work from Dominique Muller and colleagues.

- Homeostatic structural plasticity section that highlights preservation of PSD areas seems disjointed from the main focus of the article.

- Perhaps a discussion on the contribution of LTP induction paradigm/condition used on the observed effects could be useful.

- [Sustained decrease in the number of presynaptic vesicles during LTP] Could the loss of vesicles associated with loss of vesicle clusters along an axon as opposed to fusion?

- [Vesicle filling of presynaptic nascent zones enlarges the active zone during LTP] It is not clear how an immediate enlargement of the presynaptic active zone by the (fusion of) small dense core vesicles can enhance the probability of release. Would the conversion of reserve pool vesicles to primed, release ready vesicles be immediate also?

Referee #2:

This review summarized a large body of elegant and rigorous work by the Harris lab. The importance of this review is clearly reflected by the fundamental nature of the findings. Harris details pre- and postsynaptic changes in synaptic ultrastructure at various time points following LTP induction in young adult hippocampal slices. The review is well-illustrated and offers a clear snapshot of current knowledge.

My suggestions are minor and mainly focus on enhancing the readability of this review for a wider audience. In particular, some thought should be dedicated to the delivery of the findings to non-experts, with regards to definitions and use of jargon in the text.

Abstract:

- I believe it would be fair to state here that this review is primarily of own work. An early and clear distinction should also be made between adult and young adult mice, as is done later on in the main text.

- 'vesicles' should be clearly defined throughout the manuscript. An example for unclear use in the abstract: 'In hippocampal area CA1, small dense-core vesicles and tethered synaptic vesicles are recruited to presynaptic boutons enlarging active zones. By 2 hours during LTP, there is a sustained loss of vesicles...': which vesicles are lost? dense-core or synaptic vesicles?

- The abstract is composed of SV pool definitions that are not the subject of the manuscript. Instead, it would focus on the key findings that are reviewed.

Main text:

- Homeostatic structural synaptic plasticity during LTP': While the title, abstract and introduction sections primarily focus on the presynapse in LTP, this first part following the introduction is almost exclusively dedicated to the postsynapse. This transition is somewhat confusing. It would be better to define this in the section subtitle, or perhaps approach these findings from a presynaptic perspective.

- 'Changes in presynaptic boutons during LTP': the part starting from 'Related to the stalled spine outgrowth..' and until the end of the paragraph should be re-written to improve readability.

- 'In cultured hippocampal neurons, aldehyde fixation does not alter the probability of vesicular release [49].' This sentence is confusing as the fixation is expected to kill the cell. I believe the author refers to the milliseconds prior to the cessation of synaptic transmission as documented by electrophysiological recordings? please rephrase.

- As the major method used here is electron microscopy, a short definition of the methodology used to obtain data in each section and why using it is advantageous would be useful. This may help a naïve reader to understand the final paragraph ('Future tomography in large image volumes') better.

- An overview figure or a Gantt chart-like figure summarizing the LTP-induced effects at each timepoint examined at the pre- and postsynaptic compartments would be helpful to the reader.

END OF COMMENTS

Response to Review

We have prepared and submitted a visual abstract as requested.

We have modified the paper in response to review as follows.

Referee #1:

This is a perspective article from a renowned expert on synapse structure and function, focusing on the presynaptic ultrastructural changes associated with long-term potentiation (LTP) in the hippocampus. The article takes the reader through series of experiments drawn from her own work in support for enhanced presynaptic vesicle recycling during LTP. A sweeping summary of key observations is informative and of general interest. The article as it stands is somewhat convoluted and following are some suggestions to enhance accessibility.

Response: Thank you for your kind comments, and for your help in making this article better!

Comment 1: The author should clarify what the article is about at the outset. While it is mentioned as a "review", it is not a typical review of a topic but rather a review of one's own work. This comes as a surprise while reading the article.

Response: I have modified the abstract and introduction as follows:

Abstract : second sentence: "Here I provide a perspective based on ultrastructural evidence primarily from our work that supports the hypothesis for an elevated and sustained increase in the probability of vesicle release and recycling during LTP."

Introduction last sentence: Here I consolidate our ultrastructural findings, indicating that the elevation in release and recycling is sustained for at least 2 hours after the induction of LTP.

Comment 2: Relating to the point above, perhaps a broader introduction including some previous work from others that have implicated presynaptic ultrastructural changes in LTP should be included, for example, work from Dominique Muller and colleagues.

Response: I have added reference in the introduction to Rey et al., 2020 who show with photoconversion and EM analysis a dramatic increase in vesicles in the recycling pool at 30 minutes after LTP induction. I have searched, but not been able to find ultrastructural data from D. Muller and colleagues (or others) that focuses on ultrastructural presynaptic changes in vesicles. There are a couple papers regarding multi-synaptic presynaptic boutons, but there are no measures of presynaptic vesicles. Their work is primarily focused on the postsynaptic changes in dendritic spines. I would be most grateful if the reviewer could direct me to the reference(s) that he/she considers should be added. Thank you.

Comment 3: Homeostatic structural plasticity section that highlights preservation of PSD areas seems disjointed from the main focus of the article.

Response: I have retitled this section and revised the text as follows:

"Balanced postsynaptic structural plasticity during LTP."

The last two sentences were re-written to lead into the presynaptic changes as follows:

“Together these findings illustrate a homeostatic balance in total synaptic weight with LTP enlarging some synapses at the expense of new small spine outgrowth in adult hippocampal slices (Fig. 1G). The findings provide an interesting basis to discover how the presynaptic axons and vesicles respond to these profound changes in spine number and synapse size.”

Comment 4: Perhaps a discussion on the contribution of LTP induction paradigm/condition used on the observed effects could be useful.

Response: Thank you for this suggestion; I agree and I have added a section entitled “Methods for sustaining LTP and preserving synaptic ultrastructure in hippocampal slices.” The text in this new section highlights the essential features of our acute slice protocol.

Comment 5: [Sustained decrease in the number of presynaptic vesicles during LTP] Could the loss of vesicles associated with loss of vesicle clusters along an axon as opposed to fusion?

Response: I am not entirely sure what is being asked here. The drop in vesicle number is measured per bouton, not per axon length. To make that point more clear, I added the following words to the subtitle and opening paragraph:

Sustained decrease in the number of presynaptic vesicles **per bouton** during LTP

In addition to the loss of presynaptic boutons, there is a substantial drop in the number of vesicles found **in each** of the remaining boutons [22].

If I have not understood the question correctly, please let me know and I can make further revisions as needed.

Comment 6: [Vesicle filling of presynaptic nascent zones enlarges the active zone during LTP] It is not clear how an immediate enlargement of the presynaptic active zone by the (fusion of) small dense core vesicles can enhance the probability of release. Would the conversion of reserve pool vesicles to primed, release ready vesicles be immediate also?

Response: Good question. In response I have modified the perspective as follows:

- 1) I have reversed the order of presentation so that this section comes after the section on “Increased local density of tightly docked vesicles in active zones during LTP.”
- 2) I have revised the text to point out that the ultrastructural findings can only suggest that there would be an increase in the number of release sites, whether they are release ready at 5 or 30 minutes is not known as our tomograms were done only at 2 hours.
- 3) The EM tomography at 2 hours pretty strongly suggests that the shortened docking filaments enhance the release-ready pool at that time.

Referee #2:

This review summarized a large body of elegant and rigorous work by the Harris lab. The importance of this review is clearly reflected by the fundamental nature of the findings. Harris details pre- and postsynaptic changes in synaptic ultrastructure at various time points following LTP induction in young adult hippocampal slices. The review is well-illustrated and offers a clear snapshot of current knowledge.

Response: Thank you.

My suggestions are minor and mainly focus on enhancing the readability of this review for a wider audience. In particular, some thought should be dedicated to the delivery of the findings to non-experts, with regards to definitions and use of jargon in the text.

Response: Thank you for your suggestions – explanations and additions have been made as suggested below.

Comment 7: I believe it would be fair to state here that this review is primarily of own work. An early and clear distinction should also be made between adult and young adult mice, as is done later on in the main text.

Response: I have modified the abstract as indicated above for reviewer 1 as follows:

Here I provide a perspective based on ultrastructural evidence from our prior work that supports the hypothesis for an elevated and sustained increase in the probability of vesicle release and recycling during LTP.

Regarding the distinction between adult and young rats (not mice), this article is restricted to the dataset from young adult rats – I have added this distinction to the abstract: “This work is based on data collected from hippocampal area CA1 in young adult rats.” In the new methods section, I have added the age range (51–65 days old).

Comment 8: 'Vesicles' should be clearly defined throughout the manuscript. An example for unclear use in the abstract: 'In hippocampal area CA1, small dense-core vesicles and tethered synaptic vesicles are recruited to presynaptic boutons enlarging active zones. By 2 hours during LTP, there is a sustained loss of vesicles...': which vesicles are lost? dense-core or synaptic vesicles?

- The abstract is composed of SV pool definitions that are not the subject of the manuscript. Instead, it would focus on the key findings that are reviewed.

Response: I removed the SV pool definitions from the abstract. In addition, throughout the manuscript and abstract I distinguish synaptic vesicles from dense-core vesicles.

Main text:

Comment 9: 'Homeostatic structural synaptic plasticity during LTP': While the title, abstract and introduction sections primarily focus on the presynapse in LTP, this first part following the introduction is almost exclusively dedicated to the postsynapse. This transition is somewhat confusing. It would be better to define this in the section subtitle, or perhaps approach these findings from a presynaptic perspective.

Response: I have retitled this section “Balanced postsynaptic structural plasticity during LTP” and revised the text in this section to motivate the subsequent presynaptic focus.

Comment 10: 'Changes in presynaptic boutons during LTP': the part starting from 'Related to the stalled spine outgrowth..' and until the end of the paragraph should be re-written to improve readability.

Response: This section has been revised to improve readability.

Comment 11: 'In cultured hippocampal neurons, aldehyde fixation does not alter the probability of vesicular release [49].' This sentence is confusing as the fixation is expected to kill the cell. I

believe the author refers to the milliseconds prior to the cessation of synaptic transmission as documented by electrophysiological recordings? please rephrase.

Response: Fixed accordingly. Thank you.

Comment 12: As the major method used here is electron microscopy, a short definition of the methodology used to obtain data in each section and why using it is advantageous would be useful. This may help a naïve reader to understand the final paragraph ('Future tomography in large image volumes') better.

Response: I have added a section earlier in the manuscript entitled:

"Methods for sustaining LTP and preserving synaptic ultrastructure in hippocampal slices". This section outlines the critical methods needed to produce the ultrastructural quality needed for these analyses,

Comment 13: An overview figure or a Gantt chart-like figure summarizing the LTP-induced effects at each timepoint examined at the pre- and postsynaptic compartments would be helpful to the reader.

Response: We have created a visual abstract, and added a summary paragraph at the end of this perspective outlining each timepoint. Thanks for the suggestion.

Dear Professor Harris,

Re: JP-TR-2025-286983R1 "**ENHANCED CYCLING OF PRESYNAPTIC VESICLES DURING LONG-TERM POTENTIATION (LTP) IN RAT HIPPOCAMPUS**" by Kristen M. Harris

Thank you for submitting your manuscript to The Journal of Physiology. It has been assessed by a Reviewing Editor and by 2 expert referees and we are pleased to tell you that it is almost ready for acceptance. Before formal acceptance, however, we just need you to address a few administrative points - see comments below.

The review comments are copied at the end of this email.

ABSTRACT FIGURES: Authors may use The Journal's premium BioRender account to create/redraw their Abstract Figures (and any other suitable schematic figure). Information on how to access this account is here: <https://physoc.onlinelibrary.wiley.com/journal/14697793/biorender-access>.

REVISION CHECKLIST: Upload a full Response to Referees file. To create your 'Response to Referees' copy all the reports, including any comments from the Senior and Reviewing Editors, into a Microsoft Word, or similar, file and respond to each point, using font or background colour to distinguish comments and responses and upload as the required file type.

We look forward to receiving your revised submission.

Yours sincerely,

Laura Bennet

EDITOR COMMENTS

Reviewing Editor:

The author has done an excellent job of responding to previous comments.

The reviewer comment (Ref #2 below) about slowing the AP is not a big deal. You could add the word "potentially" if need be. But cable filtering properties do suggest a slower AP with larger bouton.

Please also see 'Required Items' below.

REFEREE COMMENTS

Referee #1:

My previous concerns have been suitably addressed by the author.

Referee #2:

As before, I think this is an important summary of an important body of work. My concerns have been fully addressed, and I enjoyed reading the revised version.

The only recommendation I would still make is to remove the sentence from the abstract suggesting that enlargement of the presynaptic bouton would slow the action potential. To the best of my knowledge, this has not been established experimentally.

REQUIRED ITEMS

- Please include an Abstract Figure file, **as well as the Figure Legend text within the main article file**.
It is the legend that we are missing.

- The reference list must be in alphabetical order, rather than numbered, to comply with our Journal format.

- Please upload separate high quality figure files via the submission form (one per figure, please).

END OF COMMENTS

REFEREE COMMENTS

Referee #1:

My previous concerns have been suitably addressed by the author.

Response: Thanks for all your helpful input.

Referee #2:

As before, I think this is an important summary of an important body of work. My concerns have been fully addressed, and I enjoyed reading the revised version.

Response: Thanks for all your helpful input.

The only recommendation I would still make is to remove the sentence from the abstract suggesting that enlargement of the presynaptic bouton would slow the action potential. To the best of my knowledge, this has not been established experimentally.

Response: Done as requested, and moved the text up in the abstract to follow the vesicle drop statement as underlined below. We will cite Chereau et al., 2017 on the axon velocity idea in the discussion only.

Revised text in the Abstract: By 2 hours during LTP, there is a sustained loss of vesicles, especially in presynaptic boutons containing mitochondria and clathrin-coated pits. This decrease in vesicles accompanies an enlargement of the presynaptic bouton, suggesting they supply membrane needed for the enlarged bouton surface area.

REQUIRED ITEMS

- Please include an Abstract Figure file, **as well as the Figure Legend text within the main article file**.

It is the legend that we are missing.

Added Abstract figure legend to the main article text:

Figure Legends:

Abstract Figure Legend: This visual abstract illustrates the sequence of events leading to the sustained enhancement of presynaptic vesicle cycling during LTP. At time 0, all presynaptic axonal boutons contain a pool of non-docked vesicles and docked synaptic vesicles tethered to the presynaptic active zone. A particular presynaptic axonal bouton may or may not contain a mitochondrion or small dense core vesicle. Synapses comprise nascent zones with postsynaptic

densities but no presynaptic vesicles, and active zones with both. The synaptic cleft spans both nascent and active zones. By 5 minutes after the induction of LTP, small dense core vesicles are recruited to the presynaptic membrane. Docked vesicles are reduced in number reflecting release. By 30 minutes, there are fewer vesicles overall, more coated pits, and small dense core vesicles are at their pre-LTP locations along the axons. In parallel, docked vesicles are recruited to regions of previous nascent zones converting them to active zones further enhancing the possibility of release. Two hours after the induction of LTP, the axonal bouton has enlarged, new nascent zones have appeared and are ready for new LTP. The docked vesicles are more tightly tethered and clustered at active zone release sites, suggesting a sustained elevation in presynaptic release during LTP.

Also added all other Figure Legends to main article.

- The reference list must be in alphabetical order, rather than numbered, to comply with our Journal format.

Fixed – thank you.

- Please upload separate high quality figure files via the submission form (one per figure, please)

Done

Dear Professor Harris,

Re: JP-TR-2025-286983R2 "**ENHANCED CYCLING OF PRESYNAPTIC VESICLES DURING LONG-TERM POTENTIATION (LTP) IN RAT HIPPOCAMPUS**" by Kristen M. Harris

We are pleased to tell you that your paper has been accepted for publication in The Journal of Physiology.

Authors should note that it is too late at this point to offer corrections prior to proofing. Major corrections at proof stage, such as changes to figures, will be referred to the Editors for approval before they can be incorporated. Only minor changes, such as to style and consistency, should be made at proof stage. Changes that need to be made after proof stage will usually require a formal correction notice.

Yours sincerely,

Laura Bennet
Senior Editor
The Journal of Physiology

P.S. - You can help your research get the attention it deserves! Check out Wiley's free Promotion Guide for best-practice recommendations for promoting your work at www.wileyauthors.com/eeo/guide. You can learn more about Wiley Editing Services which offers professional video, design, and writing services to create shareable video abstracts, infographics, conference posters, lay summaries, and research news stories for your research at www.wileyauthors.com/eeo/promotion.

IMPORTANT NOTICE ABOUT OPEN ACCESS: To assist authors whose funding agencies mandate public access to published research findings sooner than 12 months after publication, The Journal of Physiology allows authors to pay an Open Access (OA) fee to have their papers made freely available immediately on publication.

You can check if your funder or institution has a Wiley Open Access Account here: <https://authorservices.wiley.com/author-resources/Journal-Authors/licensing-and-open-access/open-access/author-compliance-tool.html>.